# Three-Dimensional Structure of Novel Liver Cancer Biomarker Liver Cancer-Specific Serine Protease Inhibitor Kazal (LC-SPIK) and Its Performance in Clinical Diagnosis of Hepatocellular Carcinoma (HCC)

**DOI:** 10.3390/diagnostics14070725

**Published:** 2024-03-29

**Authors:** Felix Lu, Connor Ott, Prabha Bista, Xuanyong Lu

**Affiliations:** ImCare Biotech, 3805 Old Easton Road, Doylestown, PA 18902, USA; felix.lu@imcarebiotech.com (F.L.); connor.ott@imcarebiotech.com (C.O.); prabha.bista@imcarebiotech.com (P.B.)

**Keywords:** biomarker, HCC (hepatocellular carcinoma), LC-SPIK (liver cancer-specific Serine Protease Inhibitor Kazal), AFP (alpha-fetoprotein), MASLD (metabolic dysfunction-associated steatotic liver disease)

## Abstract

LC-SPIK is a liver cancer-specific isoform of Serine Protease Inhibitor Kazal and has been proposed as a new biomarker for the detection of HCC given its unique 3D structure, which differs from normal pancreatic SPIK. An ELISA technology based on its unique structure was developed to use LC-SPIK as an effective biomarker for the clinical diagnosis of HCC. AFP, the most widely used biomarker for HCC surveillance currently, suffers from poor clinical performance, especially in the detection of early-stage HCC. In one case–control study, which included 164 HCC patients and 324 controls, LC-SPIK had an AUC of 0.87 compared to only 0.70 for AFP in distinguishing HCC from liver disease controls (cirrhosis, HBV/HCV). LC-SPIK also performed significantly better than AFP for the 81 patients with early-stage HCC (BCLC stage 0 and A), with an AUC of 0.85 compared to only 0.61 for AFP. Cirrhosis is the major risk factor for HCC; about 80% of patients with newly diagnosed HCC have preexisting cirrhosis. LC-SPIK’s clinical performance was also studied in HCC patients with viral and non-viral cirrhosis, including cirrhosis caused by metabolic dysfunction-associated steatotic liver disease (MASLD) and alcoholic liver disease (ALD). In a total of 163 viral cirrhosis patients with 93 HCC patients (50 early-stage), LC-SPIK had an AUC of 0.85, while AFP had an AUC of 0.70. For patients with early-stage HCC, LC-SPIK had a similar AUC of 0.83, while AFP had an AUC of only 0.60. For 120 patients with nonviral cirrhosis, including 62 HCC (23 early-stage) patients, LC-SPIK had an AUC of 0.84, while AFP had an AUC of only 0.72. For the 23 patients with early-stage HCC, LC-SPIK had a similar AUC of 0.83, while the AUC for AFP decreased to 0.65. All these results suggest that LC-SPIK exhibits significantly better performance in the detection of HCC than AFP in all etiologies of liver diseases. In addition, LC-SPIK accurately detected the presence of HCC in 71–91% of HCC patients with false-negative AFP test results in viral-associated HCC and non-viral-associated HCC.

## 1. Introduction

Primary HCC (hepatocellular carcinoma) is the most common and deadliest form of liver cancer, causing hundreds of thousands of deaths worldwide each year [1,2,3,4]. It usually remains undetected until its later stages, at which point patients face a 5-year survival rate of less than 15%. However, the survival rate can be over 70% if the HCC is detected early [2,5,6,7].

Currently, HCC surveillance generally includes a liver ultrasound (US) with or without AFP (alpha fetoprotein) biomarker testing [8,9,10]. US detection is noninvasive but it is operator and equipment dependent, and it is often not sensitive enough to detect HCC in its critical early stages, with an estimated sensitivity below 50% [11,12]. Moreover, US is less accurate in patients with a high BMI or those with non-alcoholic fatty liver disease (MASLD) and/or a coarse liver echotexture [13] All these factors limit the utility of ultrasound in HCC surveillance. A biomarker test that utilizes patient serum would be significantly more convenient and cheaper, but currently there are no effective HCC biomarkers available for clinical use. AFP, the most commonly used biomarker today, only has an estimated 41–65% sensitivity and 80–90% specificity for HCC [14,15]. AFP is even less accurate in detecting early-stage HCC, with a sensitivity of less than 40% [14,16]. Additionally, nearly 40% of patients with HCC have undetectable AFP levels in their sera [15,17,18,19], and patients with chronic liver diseases may have falsely elevated AFP levels during active inflammation [20]. These factors limit the application of AFP in HCC surveillance. Therefore, development of an accurate and cost-effective biomarker for HCC surveillance remains a significant unmet need. Recently, a protein called Liver-Specific Serine Protease Inhibitor Kazal (LC-SPIK), which is secreted specifically by liver cancer cells in blood, was identified. LC-SPIK is a liver cancer-specific isoform of Serine Protease Inhibitor Kazal (SPIK/SPINK). Here, we will evaluate the use of LC-SPIK as a new biomarker for detection of HCC.

## 2. SPIK and the Development of Cancer

SPIK is a small protein with 79 amino acids; it is also called PSTI (pancreas secretory trypsin inhibitor) and TATI (tumor-associated trypsin inhibitor) [21,22]. Normally, SPIK has no or limited activity in liver tissues or any tissues besides the pancreas. Numerous studies have reported that the expression of SPIK may be elevated in cancer such as HCC [23,24,25,26,27,28,29] and that high expression of SPIK is closely related to the progression of HCC [25,29]. 

The reasons for this over-expression of SPIK triggering cancer development have been the subject of research. The most compelling evidence comes from the studies of the function of SPIK that describe SPIK as activated as a reactant during inflammation [30,31,32,33]. For example, SPIK was activated in rat liver cells to counter turpentine-induced liver inflammation [33]. SPIK was also activated in response to inflammatory cytokines during human viral hepatitis [34]. Lamontagne et al. showed that replication of the hepatitis B virus and hepatitis C virus, two main causes of chronic hepatitis, can up-regulate the expression of SPIK [35]. Interestingly, a high number of SPIK transcripts was correlated with cancer progression and recurrence after surgical resection [24,25,36]. Furthermore, the highest levels of SPIK are often associated with the latest stages of cancer, probably implying a cumulative, dose-dependent effect of SPIK on cell transformation [37]. Together, these studies suggest that in addition to, or perhaps because of, its role as an inflammatory protein, SPIK may play an important role in the formation and development of cancer [26]. 

The progression of cancer could be due to, at least in part, the tolerance of cancer cells to the body’s immune surveillance, in other words, the evasion of the body’s immune response and immune-mediated clearance. This results in the body’s inability to induce cell death in abnormal cells, resulting in uncontrolled cell growth progressing into cancer (Figure 1) [38,39]. Generally, in immune surveillance, cytotoxic T lymphocytes (CTLs) and natural killer (NK) cells secrete apoptotic cytolytic granules such as granzyme A (GzmA) and granzyme B (GzmB), which initiate the apoptotic pathway in target cells. This occurs with help from perforin, a protein that triggers pore formation in the cellular membrane of target cells and allows GzmA/B to enter. The clearance of abnormal cells by immune surveillance maintains the body in a healthy state (Figure 1: Normal). Because both GzmA and GzmB are cytotoxic serine proteases, it is possible that the GzmA/GzmB-induced apoptosis may be blocked by increased expression of a protease inhibitor. Over-expression of a protease inhibitor could ontogenetically impact cell proliferation, resulting in the abnormal cells evading immune killing induced by CTLs and NK cells, allowing uncontrolled growth of abnormal cells and leading to the development of cancer (Figure 1: Cancer) [40,41]. Because SPIK is a serine proteinase inhibitor and GzmA/B are serine proteinases, it is viable that SPIK interacts with GzmA/B, preventing them from initiating cell apoptosis in abnormal cells and resulting in their escape from immune clearance [42,43,44]. The uncontrolled abnormal cells then further develop to cancer (Figure 1: Cancer). The inhibition of GzmA by SPIK is supported by the observation that rat SPIK could bind to GzmA and inhibit its ability to hydrolyze substrates such as N-a-benzyloxycarbonyl L-lysine thiobenzyl ester [45]. Lu et al. also demonstrated that anti-GzmA antibody could co-immunoprecipitate SPIK and suppress GzmA-induced serine protease-dependent cell apoptosis (SPDCA) in cell culture [26]. Pardo and Lieberman found that at low nanomolar concentrations, GzmA triggered a pro-inflammatory effect, whereas at high nanomolar concentrations, GzmA induced SPDCA [42,46]. SPIK was also reported to be able to inhibit GzmB-induced apoptosis [47,48]. However, the apoptosis induced by GzmB is caspase-dependent apoptosis (CDA), which is a different apoptotic pathway than SPDCA [49,50,51]. It is likely that suppression of GzmA/GzmB-induced apoptosis, including both SPDCA and CDA, by the over-expression of SPIK, would eventually result in the escape of liver cancer cells from immune clearance and even suppress the immune response [42,43]. This hypothesis is further confirmed by the observation that high levels of SPIK are closely associated with the early recurrence of HCC in patients following surgical resection [24,25]. Because recurrence of cancer often implies an inability of the immune system to clear lingering oncogenic cells, early recurrence of HCC in patients with high levels of SPIK raises the possibility that the over-expression of SPIK interferes with the elimination of lingering oncogenic cells by the immune system. The uncontrolled expansion of these lingering cells triggers cancer recurrence.

## 3. The Difference between LC-SPIK and Normal SPIK

The use of SPIK as a HCC biomarker was explored after it was discovered as a protein secreted into patients’ blood [44,52]. However, the use of SPIK as a cancer biomarker is impeded by the fact that serum levels of SPIK are also elevated in the presence of other diseases, especially pancreatitis [53,54,55]. Lu et al. found that although all cells, including HCC cells, express identical SPIK in the cytoplasmic form, the secreted form of SPIK is unique to HCC cells. For pancreatic SPIK (pan-SPIK) or normal SPIK produced by the pancreas, a 23 amino-acid fragment in the N-terminus, assumed to be a signal peptide, is removed during secretion, but for SPIK secreted by liver cancer cell lines or HCC, this segment is retained [26,56,57]. We call the SPIK secreted from liver cancer cells the Liver Cancer SPIK (LC-SPIK). The sequence of amino acids of LC-SPIK and SPIK is listed in Figure 2A. The extra 23 AA sequence, which only exists in LC-SPIK, is underlined. The common region of LC-SPIK and SPIK (AA sequence Nos. 24–79) is also listed (Figure 2A). The size differences between LC-SPIK secreted from liver cancer cells and normal SPIK secreted from pancreatic cells were compared by Western blot testing with monoclonal antibodies IMCA1 and MA86, which bind to the 23 extra AA fragment and the common region shared by LC-SPIK and pan-SPIK, respectively. The culture medium from a liver cancer cell line (S2–3) and a pancreatic cell line (PanC1) was collected and analyzed by the Western blot technique. Figure 2B shows that only LC-SPIK from S2–3 cells, not SPIK from pancreatic cells (pan-SPIK), was recognized by antibody IMCA1, suggesting it has extra 23 AA in the N-terminus. In contrast, both SPIKs were recognized by MA86, which suggests that it binds to the common region shared by LC-SPIK and pan-SPIK (Figure 2B, MA86). The size of the S2–3 generated protein (LC-SPIK) was around 8.5 kDa, which corresponds to the correct molecular weight for a full-length genetic SPIK, confirming that the LC-SPIK secreted by cancer cells has an entire sequence of SPIK [26,35,58]. However, the secreted protein from the PanC1 cell line (pan-SPIK) showed that the size of the protein is smaller than its counterpart from the cell lines S2–3 (Figure 2B MA86), at around 6 kDa. This corresponds to an attenuated SPIK with 56 amino acids, suggesting that pancreatic SPIK is proteolytically cleaved upon secretion [21,59]. Edman N-terminal analysis further confirmed this conclusion (Figure 2C). The LC-SPIK secreted by liver cancer cells has amino acids VTG in positions 2–4, confirming that LC-SPIK has an entire sequence of genetic SPIK (Figure 2C, Edman N-terminal analysis). The quantity of LC-SPIK in HCC patients’ serum was also much greater than the quantity of SPIK secreted by pancreatic cells. This was confirmed by analysis of serum samples from patients with HCC with Western blot testing. The results showed that the SPIK with molecular weight around 8.5 KDa existed in the serum of all six examined patients, which is the same molecular weight of LC-SPIK from S2–3, while the SPIK in patients with pancreatitis had a small molecular of 6 kDa [26,60,61]. The reason cancer cells can secrete unattenuated SPIK is unclear. Considering SPIK is a serine proteinase inhibitor and signal peptidase is a serine proteinase, it is possible that over-expression of SPIK inhibits signal peptidase activity, resulting in the secretion of an entire uncut protein.

## 4. 3-D Structure of LC-SPIK

The extra 23 amino-acid fragments in the N-terminus of LC-SPIK not only change the length of the protein but potentially also change the whole protein configuration. Three-dimensional crystal structure analysis of LC-SPIK and SPIK suggests that the presence of these additional 23 residues in the N-terminus of LC-SPIK causes it to have a different conformation compared to normal pan-SPIK. Compared to the 3D structure of normal SPIK as reported by Hecht et al. [56] the crystal structure of LC-SPIK determined by the CLIPS protein epitope mapping study (Pepscan, Lelystad, The Netherlands) has obvious differences, both in conformation and configuration. Through a visual comparison, three conformational differences between LC-SPIK and SPIK can be identified, which are outlined in red boxes in Figure 3. Box I shows the N-terminus of both SPIK and LC-SPIK. The extra 23-residue fragment in LC-SPIK projects outwards and extends past the main body of the protein; in contrast, the N-terminus of SPIK does not have this additional fragment. This exposed fragment greatly increases the likelihood of other proteins and antibodies selectively interacting with LC-SPIK but not SPIK. Box II shows that, due to the longer N-terminus of LC-SPIK, the first loop in LC-SPIK is flatter and angled differently compared to the corresponding loop in SPIK. This difference leads to more space between the first loop and the alpha helix (Figure 3, box II) in LC-SPIK, which exposes amino acids that are on the interior and inaccessible in SPIK. Finally, as shown in Box III, the longer N-terminus of LC-SPIK also changes the relative position and distance between the N-terminus and the alpha-helix of the protein as well as the loop after it. The loop in SPIK nearly disappears in LC-SPIK. These three changes and differences in tertiary structure suggest that LC-SPIK may have a different conformation compared to normal SPIK and that conformation-dependent antibodies could be generated to specifically target either form of SPIK. 

## 5. Development of Anti-LC-SPIK Antibody and Test Kit

Based on the structural difference between LC-SPIK and SPIK, it is possible to develop an antibody that would recognize and bind specifically to LC-SPIK, which is secreted by cancerous liver cells, allowing us to differentiate HCC from non-cancerous liver disease [60]. Indeed, we successfully developed a monoclonal anti-LC-SPIK antibody MCA, which specifically binds to LC-SPIK but not SPIK [61]. Our epitope analysis shows that MCA is a conformation-dependent antibody, and the epitope it binds to specifically is composed of two discontinuous fragments. One fragment is located within the 23 N-terminal residues, which are removed from SPIK during secretion, while the second is within the common region shared by both LC-SPIK and SPIK (Figure 4 and Figure 5A). The 3D structure of the epitopes and antibody binding sites is shown in Figure 4. 

Using this specific anti-LC-SPIK antibody, we further developed an ELISA test kit [61]. This would support the development of a diagnostic technology that can selectively and reliably detect HCC without interference from other liver or non-liver diseases, such as liver cirrhosis (viral and non-viral), hepatitis, and pancreatitis. Figure 5B shows that MCA binds specifically to LC-SPIK but not pancreatic SPIK, while Poly A, a polyclonal anti-SPIK antibody that binds the common region of SPIK, recognizes both kinds of SPIK.

## 6. LC-SPIK and AFP Expression in Serum of Patients with HCC

Using this test kit, we evaluated the ability of LC-SPIK to differentiate between patients with primary HCC and non-cancerous liver disease. At the same time, we evaluated its clinical performance against AFP. A total 488 patients participated in the study, including 164 patients with HCC and 324 controls without HCC [61]. Of the 164 HCC patients, 81 were considered early-stage HCC (BCLC stage 0-A) and 83 were considered late-stage (BCLC stage B-D). Of the 324 controls, 245 were non-cancer liver disease and 79 were healthy patients used to establish a baseline. Among the 245 liver diseases patients, 125 had liver cirrhosis of various etiologies and 120 had chronic HBV/HCV infection without cirrhosis. The results showed LC-SPIK can distinguish HCC patients from controls with an area under the curve (AUC) of 0.87 (95% CI: 0.84 to 0.91), as well as 80% sensitivity and 90% specificity using a cut-off value of 21.5 ng/mL. AFP had an AUC of 0.70 (95% CI: 0.64 to 0.76), with 52% sensitivity and 86% specificity using a cut-off value of 20.0 ng/mL (Table 1A) [61]. The difference in AUC between LC-SPIK and AFP was 0.17 (*p* < 0.001), suggesting LC-SPIK performed significantly better as an HCC biomarker than AFP.

LC-SPIK also performed well in its ability to detect early-stage HCC. In 81 patients with early-stage HCC, the AUC of LC-SPIK was 0.85, with slightly decreased sensitivity of 72% and the same specificity of 90%. This is significantly higher than the AUC of AFP, which was only 0.61, with 42% sensitivity and 86% specificity [61]. The difference in AUC between LC-SPIK and AFP in detecting early-stage HCC increased from 0.17 to 0.24 (*p* < 0.001), suggesting that there is an even larger performance difference between LC-SPIK and AFP for early-stage HCC (Table 1B).

Because LC-SPIK was found in patients with early-stage and even very early-stage HCC (BCLC stage 0), [61] it would be interesting if serum levels of LC-SPIK correlated with cancer progression. Lu et al. reported that mean values of LC-SPIK were consistently higher for later and more advanced stages [61]. However, due to limited samples, especially the low numbers of BCLC stage 0 (very early) and stage D (terminal) groups, their study results were not statistically significant (*p* > 0.05) [61]. If true, LC-SPIK not only might be a biomarker for the detection of early-stage HCC but also might be useful as a tool for monitoring treatment efficacy or an indicator for recurrence of HCC after resection. Additional research is needed to investigate these possibilities.

## 7. Rise of Non-Viral Risk Factors for HCC

Cirrhosis is the most significant risk factor for HCC; about 80% of patients with newly diagnosed HCC have pre-existing cirrhosis [62,63]. Generally, cirrhosis falls into two separate categories: (1) viral cirrhosis (e.g., also has chronic HBV or HCV) and (2) non-viral cirrhosis, which includes alcoholic liver disease (ALD), metabolic dysfunction-associated steatotic liver disease (MASLD), and metabolic dysfunction-associated steatohepatitis (MASH). Recently, the prevalence of cirrhosis has been on the rise due to non-viral risk factors, especially due to obesity-related MASLD and alcoholic liver disease, and this has led to a corresponding rise in related HCC [62,64,65]. These HCC patients have poor treatment outcomes, which is potentially related to altered US visualization from the presence of subcutaneous fatty tissue in addition to hepatic steatosis, consequently leading to under-recognition of small or early-stage HCC nodules [66,67]. This increases the challenges around US examination and lowers its efficacy as a surveillance tool for early-stage HCC for these patients. Adding AFP measurement to US examination can increase the sensitivity for detecting early-stage HCC but does not obviously improve the surveillance results [62,68,69,70]. To evaluate if LC-SPIK can be an equally effective biomarker in patients with non-viral cirrhosis, LC-SPIK levels in both viral and non-viral cirrhosis patients were evaluated and LC-SPIK and AFP performance was evaluated to detect HCC in viral cirrhosis.

In our study mentioned above, we also examined the serum levels of LC-SPIK and AFP in a total of 163 patients with viral cirrhosis, including 93 patients with HCC and 70 without HCC. Of the 93 HCC patients, 50 had early-stage HCC. The data are shown in Table 2. The AUC of LC-SPIK in detection of all HCC was 0.85, with a sensitivity and specificity of 81% and 89%, respectively, when using a cut-off of 21.5 ng/mL. At the same time, the AUC of AFP in these patients was 0.70, with a sensitivity and specificity of 55% and 74%, respectively, when using a cutoff of 20 ng/mL [1] (Table 2A). The difference in AUC between LC-SPIK and AFP was 0.15 (*p* < 0.001), suggesting that LC-SPIK was a better biomarker than AFP in detecting HCC in patients with viral cirrhosis. For early-stage HCC, the AUC of LC-SPIK in detecting HCC was 0.83, with a sensitivity and specificity of 76% and 89%, respectively. At the same time, the AUC of AFP was 0.60, with a sensitivity and specificity of 44% and 74%, respectively (Table 2B). The AUC difference between LC-SPIK and AFP in the detection of early-stage HCC was 0.23 (*p* < 0.001). 

## 8. LC-SPIK and AFP Performance in Detecting HCC Due to Non-Viral Cirrhosis

Recently, Caviglia et al. examined the level of LC-SPIK in a total of 120 patients with non-viral cirrhosis, including 62 patients with HCC and 58 without HCC [71]. Of the 62 HCC patients, 23 had early-stage HCC. The AUC of LC-SPIK in detection of HCC in their test was 0.84, and the sensitivity and specificity was 89% and 66%, respectively. In the same samples, the AUC of AFP in these patients was 0.72, with a sensitivity and specificity of 70% and 60%, respectively (Table 3A). The difference in AUC between LC-SPIK and AFP was 0.12 (*p* < 0.001). For the detection of early-stage HCC, the performance of LC-SPIK did not diminish and had an AUC of 0.83 with the same sensitivity and specificity of 89% and 66%, respectively. AFP, however, saw its AUC decrease significantly from 0.72 to 0.65, and the sensitivity was reduced from 70% to 59% (Table 3B). The obviously increase of AUC of LC-SPIK compared to AFP (0.84 vs. 0.72), especially in detecting early-stage HCC (0.83 vs. 0.65), is significant because AFP has demonstrated poor performance as a biomarker for HCC in non-viral cirrhosis patients, particularly for early-stage HCC [71]. 

Compared to patients with other liver diseases, cirrhotic patients had a higher false-positive rate of LC-SPIK (18.4%) [61], which might be related to the fact that they also have the highest risk of developing HCC. There is a possibility that LC-SPIK, in some of these false-positive cases, may be detecting very early-stage HCC that was missed by surveillance imaging. However, this possibility and sensitivity compared to imaging must be evaluated further.

## 9. Detection of HCC in Patients with False-Negative AFP Test Results

As mentioned before, about 40% of HCC patients have serum AFP results that are considered negative, which greatly limits the use of AFP as an effective biomarker in HCC surveillance [11,12,15,18]. Therefore, there is a significant unmet need for an effective diagnostic tool for HCC in these patients, where AFP is not effective. LC-SPIK’s role in detecting HCC in AFP false-negative patients of different etiologies was studied. Table 4 shows that 77 of 164 HCC patients (Table 1A) had AFP-negative results; thus, the false negative rate is 47%. Of these 77 AFP-negative patients, 55 had true-positive LC-SPIK results, with an accurate diagnosis rate of 71% [61]. The AUC was 0.78, suggesting the significant high sensitivity and specificity of LC-SPIK in detection of HCC in these patients. Similar results were obtained when looking at HCC patients with viral cirrhosis (Table 2A) and non-viral cirrhosis (Table 3A). Table 4 shows that 42 of 93 viral HCC patients were AFP negative, with a false-negative rate of 45%. Of these 42 patients, 35 of them had positive LC-SPIK results, with an accurate diagnosis rate of 83%. The AUC was 0.81. Among HCC patients with nonviral cirrhosis, 23 out of 62 were identified as AFP false negative (37%); of these, LC-SPIK was positive in 21 patients (accurate diagnosis rate of 91%) (Table 4). The AUC was high, at 0.91 [71]. Similar results were observed in the detection of AFP-negative patients with early-stage HCC. Overall, this shows that LC-SPIK is especially sensitive in patients where AFP is negative and does not have any significant differences between viral and non-viral HCC.

## 10. Combination of LC-SPIK Test with Other Biomarkers in Diagnosis of HCC

In order to improve the overall clinical performance of the LC-SPIK test, LC-SPIK was combined with other HCC biomarkers, such as AFP and PIVKA-II (also known as DCP), other proteomic biomarker for HCC with relatively strong support in the clinical literature [11,72,73]. Caviglia et al. reported that combining LC-SPIK with AFP and/or PIVKA-II can improve overall diagnostic performance significantly. In a study with 96 patients, including 58 with cirrhosis and 38 with early-stage HCC, the combination of LC-SPIK and AFP increased the AUC of the LC-SPIK test from 0.841 to 0.897, an increase of 0.056; for AFP, the AUC increased from 0.719 to 0.897, an increase of 0.178 (Table 5, LC-SPIK + AFP). If the combination of LC-SPIK with PIVKA-II is used, the AUC increases from 0.841 to 0.926, an increase of 0.085; for PIVKA-II, the AUC increases from 0.853 to 0.926, an increase of 0.073. If you combine all three biomarkers, the AUC of the test increases to 0.932; comparing LC-SPIK alone, it increases to 0.091 (Table 5) [71]. Our study also showed that if LC-SPIK is combined with AFP, the AUC of detection of HCC in all etiologies increased from 0.87 to 0.92. Of course, there is potential to further improve LC-SPIK’s performance through artificial intelligence or by combining LC-SPIK with other proteomic biomarkers, DNA/RNA from exosomes, cell-free DNA, circulating tumor DNA, and various serum RNAs [74]. 

## 11. Summary

As a novel HCC biomarker, LC-SPIK could be an asset in the management of HCC, especially when used to detect early-stage HCC. Compared with AFP, LC-SPIK shows better performance in the detection of HCC in all stages and etiologies and is especially useful in cases where AFP would provide a false-negative result. Because LC-SPIK represents a new biomarker for additional testing, large-scale prospective studies are required to evaluate its performance in patients with various etiologies. In addition, we believe that there is significant potential for LC-SPIK to be combined with other biomarkers, such as AFP and PIVKA-II, other genomic biomarkers, or AI, to greatly improve clinical performance and significantly improve clinicians’ ability to detect and manage HCC.

## Figures and Tables

**Figure 1 diagnostics-14-00725-f001:**
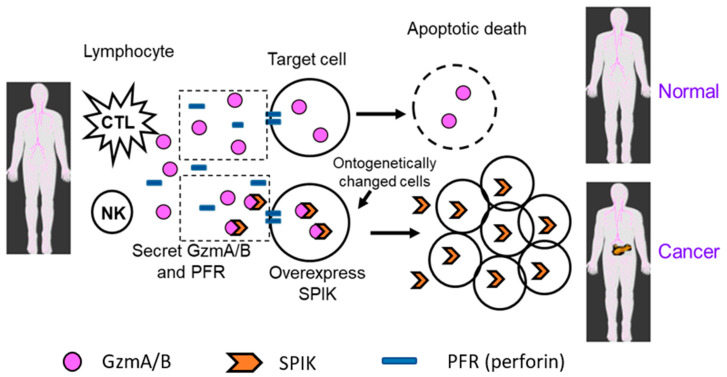
The possible relationship of SPIK and cancer development. In the immune system, when CTL and NK cells find abnormal cells, they secrete cytolytic granules such as GzmA and B, which trigger target cell apoptosis with the help of perforin, resulting in a hole in the cell member and allowing GzmA/B to enter. In a normal situation (normal), this will remove abnormal cells and keep the body healthy. However, if the abnormal cell over-expresses SPIK, a serine proteinase inhibitor that is able to inhibit the activity of GzmA and B because both are serine proteinases, this will result in the escape of abnormal cells from immune killing and uncontrolled growth, resulting in cancer (cancer).

**Figure 2 diagnostics-14-00725-f002:**
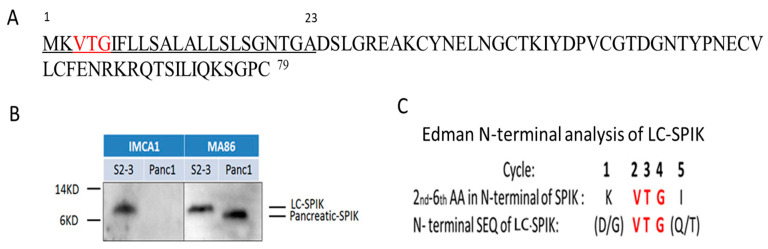
The size of LC-SPIK secreted by HCC. (**A**) Entire sequence of LC-SPIK. Compared to SPIK, LC-SPIK has an extra 23 amino acids in the N-terminus, which are underlined. (**B**) Medium from HCC-derived cell lines S2–3 and human pancreatic cells (Panc-1) as run on SDS-PAGE gel. After transfer to a PVDF membrane, proteins were visualized by staining with monoclonal antibodies IMCA1 and MA86, which bind an extra 23 AA in the N-terminus of LC-SPIK and the common region of LC-SPIK and SPIK (From numbers 24–79), respectively, as determined by Western blotting. (**C**) Edman N-terminal analysis of LC-SPIK secreted from liver cancer cell line S2–3. The sequence predicted by Edman degradation in the N-terminal of LC-SPIK is red [2,26].

**Figure 3 diagnostics-14-00725-f003:**
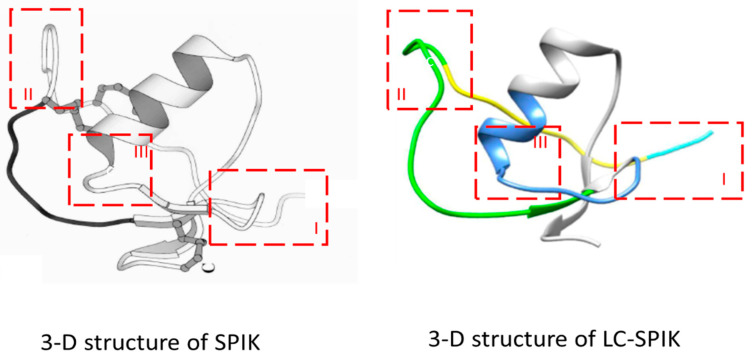
The comparison of the 3D structures of LC-SPIK and normal SPIK. The light blue indicates the extra 23 AA in LC-SPIK, which are depleted in SPIK during secretion. The red boxes I, II, and III indicate clear conformational differences between pancreatic SPIK and LC-SPIK.

**Figure 4 diagnostics-14-00725-f004:**
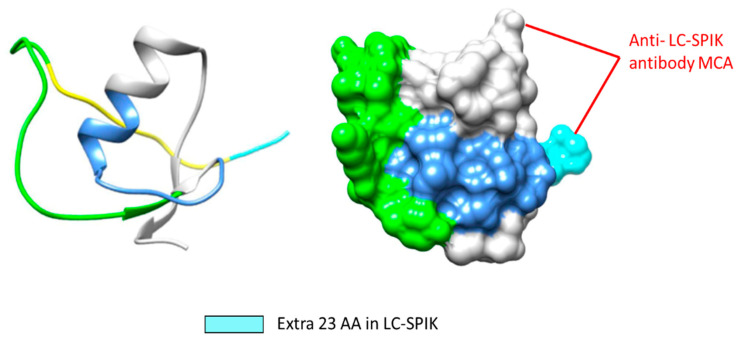
The possible epitope to which the anti-LC-SPIK antibody MCA binds.

**Figure 5 diagnostics-14-00725-f005:**
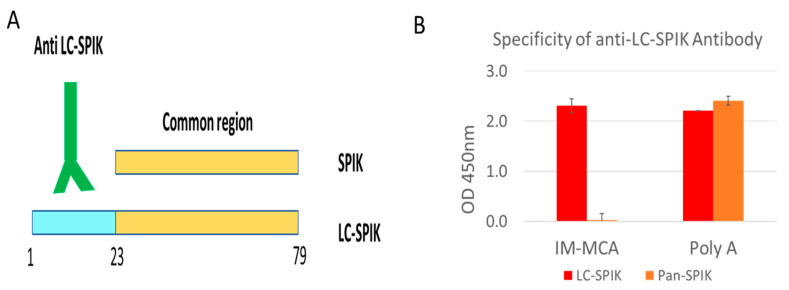
(**A**) Diagram of the special sequence of LC-SPIK and the working mechanism of anti-LC-SPIK antibody. (**B**) Anti-LC-SPIK antibody IM-MCA only recognizes LC-SPIK, not SPIK from the pancreas, while polyclonal anti-SPIK recognizes both LC-SPIK and pan-SPIK.

**Table 1 diagnostics-14-00725-t001:** Performance of LC-SPIK and AFP in detection of HCC.

**A**					
HCC vs. liver disease	AUROC area	Sensitivity	Specificity		
LC-SPIK in HCC	0.87	80%	90%	HCC	164
AFP in HCC	0.70	52%	86%	Control	324
**B**					
HCC vs. liver disease	AUROC area	Sensitivity	Specificity		
LC-SPIK in early HCC	0.85	72%	90%	HCC	81
AFP in early HCC	0.61	42%	86%	Control	324

Lu et al. [61].

**Table 2 diagnostics-14-00725-t002:** LC-SPIK and AFP in viral cirrhosis patients.

**A**					
HCC in viral cirrhosis	AUROC area	Sensitivity	Specificity		
LC-SPIK in HCC	0.85	81%	89%	Viral HCC	93
AFP in HCC	0.70	55%	74%	Viral cirrhosis	70
**B**					
HCC in viral cirrhosis	AUROC area	Sensitivity	Specificity		
LC-SPIK in early HCC	0.83	76%	89%	Viral HCC	50
AFP in early HCC	0.60	44%	74%	Viral cirrhosis	70

Lu et al. [61].

**Table 3 diagnostics-14-00725-t003:** LC-SPIK and AFP in non-viral cirrhosis patients.

**A**					
HCC in non-viral cirrhosis	AUROC area	Sensitivity	Specificity		
LC-SPIK in HCC	0.84	89%	66%	non-viral HCC	62
AFP in HCC	0.72	70%	60%	non-viral cirrhosis	58
**B**					
HCC in non-viral cirrhosis	AUROC area	Sensitivity	Specificity		
LC-SPIK in early HCC	0.83	89%	66%	non-viral early HCC	23
AFP in early HCC	0.65	59%	62%	non-viral cirrhosis	58

Caviglia et al. [71].

**Table 4 diagnostics-14-00725-t004:** Accurate diagnosis of HCC in AFP false-negative patients.

Patients with	Total Case	Negative AFP	False-Negative Rate	Negative AFP, Positive LC-SPIK	AUC for LC-SPIK (In AFP Negative HCC Patients)	Accurate Diagnosis Rate
HCC *	164	77	47%	55	0.78	71%
Viral HCC *	93	42	45%	35	0.81	83%
Non-viral HCC **	62	23	37%	21	0.91	91%

* Lu et al. [61]; ** Caviglia et al. [71].

**Table 5 diagnostics-14-00725-t005:** AUC of multi-biomarker test.

Marker	AUC Alone	LC-SPIK + AFP		LC-SPIK + PIVKA-II		LC-SPIK + AFP + PIVKA-II	
		AUC	AUC Increase	AUC	AUC Increase	AUC	AUC Increase
LC-SPIK	0.841	0.897	0.056	0.926	0.085	0.932	0.091
AFP	0.719		0.178		X		0.213
PIVKA-II	0.853		X		0.073		0.079

Caviglia et al. [71].

## Data Availability

Not applicable.

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
