# Peer review of "Three-Dimensional Structure of Novel Liver Cancer Biomarker Liver Cancer-Specific Serine Protease Inhibitor Kazal (LC-SPIK) and Its Performance in Clinical Diagnosis of Hepatocellular Carcinoma (HCC)"

_diagnostics, 2024, doi:10.3390/diagnostics14070725_

Round 1

Reviewer 1 Report

Comments and Suggestions for Authors

In this review by Felix Lu et al., the authors describe the role of the liver cancer-specific isoform of Serine Protease Inhibitor Kazal (LC-SPIK) as a possible marker for hepatic tumors (HCC). They describe SPIK and the development of cancer and then they proceed with the description of differences between normal SPIK (pancreatic form) and LC-SPIK (liver form) as well as its 3D conformation, describing the importance of an antibody capable of discriminate between the 2 forms (SPIK vs LC-SPIK), describing how this tool has higher sensitivity and specificity than AFP in the diagnosis of HCC and early HCC.

This review is interesting and highlights the importance of finding new markers for the prompt diagnosis of HCC. However, some points need to be better discussed/presented: e.g. they present results in figure 2 but they do not state the reference for these results. In the paragraph 4, lines 203-204, they state that they "successfully developed a monoclonal anti-LC-SPIK anibody"  and also an "ELISA test kit" (line 2016) but they do not refer to any reference and/or study supporting these statements. 

They never describe the "normal range" of LC-SPICK or when it is considered as diagnostic marker. Is there a possibility of false positive results? Is it possible that LC-SPIK is produced and/or highly expressed in particular situations such as metabolic diseases?

Please check for minor typo (e.g. page 4 line 133 "animo acid").

Author Response

1. They present results in figure 2 but they do not state the reference for these results.

Response: added.

2. In the paragraph 4, lines 203-204, they state that they "successfully developed a monoclonal anti-LC-SPIK antibody” and also an "ELISA test kit" (line 2016) but they do not refer to any reference and/or study supporting these statements.

Response: added.

3. They never describe the "normal range" of LC-SPICK or when it is considered as diagnostic marker.

Response: We have stated in paper line 229 “the cut-off value of LC-SPIK is 21.5ng/ml,” that implies if LC-SPIK level in patients’ serum is larger than 21.5ng/ml (positive), the result suggests patients having higher risk of HCC.

4. Is there a possibility of false positive results? Is it possible that LC-SPIK is produced and/or highly expressed in particular situations such as metabolic diseases?

Response: the related content was added as the reviewer suggested (line 240-248 in revised version).

5. Please check for minor typo (e.g. page 4 line 133 "animo acid").

Response: corrected.

Reviewer 2 Report

Comments and Suggestions for Authors

I congratulate the authors for their manuscript on LC-SPIK and its use for diagnosing HCC. The paper is well-written and uses the available evidence published in the literature. 

I have only minor comments: 

Please replace the NAFLD acronym with the most recent MASLD per international guidelines (Rinnella, J Hepatol, 2023). 

Please add some information on the use of LC-SPIK to guide the evaluation of treatment response. There are some hints at the role of LC-SPIK for the detection of HCC recurrence after resection, but It would be interesting to get information (or speculations) about SPIK and treatment efficacy.  

Author Response

1. Please replace the NAFLD acronym with the most recent MASLD per international guidelines (Rinnella, J Hepatol, 2023).

Response; updated.

2. Please add some information on the use of LC-SPIK to guide the evaluation of treatment response. There are some hints at the role of LC-SPIK for the detection of HCC recurrence after resection, but it would be interesting to get information (or speculations) about SPIK and treatment efficacy.

  Response: the related content was added as the reviewer suggested (line 294-299 in revised version).